# Finite-temperature critical behavior
# of long-range quantum Ising models

E. Gonzalez Lazo[1,2*], M. Heyl[3], M. Dalmonte[1,2], A. Angelone[1,2]

**1** SISSA, via Bonomea 265, 34136 Trieste, Italy
**2** The Abdus Salam International Centre for Theoretical Physics, Strada Costiera 11, 34151 Trieste, Italy
**3** Max Planck Institute for the Physics of Complex Systems, Nöthnitzer Straße 38, Dresden 01187, Germany
* egonzale@sissa.it

May 17, 2021

## 1 Abstract

We study the phase diagram and critical properties of quantum Ising chains with long-range ferromagnetic interactions decaying in a power-law fashion with exponent $\alpha$, in regimes of direct interest for current trapped ion experiments. Using large-scale path integral Monte Carlo simulations, we investigate both the ground-state and the nonzero-temperature regimes. We identify the phase boundary of the ferromagnetic phase and obtain accurate estimates for the ferromagnetic-paramagnetic transition temperatures. We further determine the critical exponents of the respective transitions. Our results are in agreement with existing predictions for interaction exponents $\alpha > 1$ up to small deviations in some critical exponents. We also address the elusive regime $\alpha < 1$, where we find that the universality class of both the ground-state and nonzero-temperature transition is consistent with the mean-field limit at $\alpha = 0$. Our work not only contributes to the understanding of the equilibrium properties of long-range interacting quantum Ising models, but can also be important for addressing fundamental dynamical aspects, such as issues concerning the open question of thermalization in such models.

## 1  Introduction

Systems featuring long-range interactions are central in condensed matter and statistical physics, due to both their widespread presence in nature and the wide range of characteristic physical phenomena they display, the latter often being at odds with well-known predictions and results concerning short-range models (see, e.g, [1] for a review). Within the last decade, the interest in quantum long-range interacting models has further surged due to the progress in manipulating and controlling these systems at an unprecedented level [2–6]. Specifically, these experimental platforms naturally realize long-range quantum Ising or Heisenberg models, with the possibility to engineer many-body interaction potentials decaying proportionally to $d^{-\alpha}$ as a function of distance $d$, ranging from van-der-Waals-like ($\alpha = 6$) and dipolar interactions ($\alpha = 3$) in the context of Rydberg atoms [3,6], to Coulomb ($\alpha = 1$) and infinite-range ($\alpha = 0$) potentials for trapped ions [2,5].

Recent experiments in such long-range interacting models have mostly centered on the investigation of inherent dynamical phenomena, such as many-body localization [7], discrete time crystals [8,9], prethermalization [10], Kibble-Zurek mechanism [11,12], or dynamical quantum phase transitions [13,14]. Despite of recent progress [15,16] one key question has, however, remained open: especially in the limit of small interaction exponents, it is not known whether these long-range systems follow the fundamental principle of thermalization as expected for generic short-range models. In the first place, this obviously requires a thorough understanding of the thermal properties of the system of interest, which have only been partially explored even in paradigmatic Hamiltonians such as the one-dimensional long-range quantum Ising model.

In particular, the ground-state properties of the latter in the case of ferromagnetic interactions have been the focus of investigation via analytical and renormalization group (RG) techniques [17,18], as well as linked-cluster expansions [19], tensor network approaches and/or density matrix RG [20,21], Monte Carlo methods [22] and, very recently, Stochastic Series Expansion (SSE) Monte Carlo [23] investigation in the $\alpha > 1$ region, demonstrating, e.g., that the critical behavior of the model belongs to the mean-field and short-range universality class (UC) for $1 < \alpha < 5/3$ and $\alpha \geq 3$, respectively. The antiferromagnetic case has also been intensely studied via the use of several approaches [19,23–27], with notable results including, among others, the demonstration that the half-chain entanglement entropy displays area-law violations in the intermediate regime $1 < \alpha < 2$ [24]. Considerable effort has also been dedicated to the theoretical investigation of the dynamical properties of this type of model [28–34].

Oppositely with respect to the zero-temperature case, the finite-temperature regime is still poorly understood. Indeed, the latter has been predicted by general theoretical arguments [35] to belong to the universality class of the corresponding classical long-range Ising model, with quantum effects not changing this description at the qualitative level. While this picture has been essentially confirmed for the case $\alpha = 3$ by SSE studies [36], the latter demonstrated, in the proximity of the ground-state critical point, the presence of considerable finite-size effects induced by strong quantum fluctuations, which all but prevent observation of the expected classical regime even at very large system sizes.

In the light of the experimental realizations of these models discussed above, investigating the thermal critical behavior of these Hamiltonians remains therefore of great

importance, in order to determine the role and strength of the quantum effects in perturbing the predicted classical picture. Furthermore, (numerically) exact analysis of the finite-temperature regime is essential to determine non-universal details such as, e.g., the position of thermal critical points, which are influenced in a key way by quantum effects, and whose knowledge is crucial for laboratory realizations. Such a study is of especially great interest in the extremely long-ranged regime $0 < \alpha < 1$, which, to our knowledge, has not been the object of this kind of investigation, and (as mentioned above) is directly realizable in trapped-ions setups.

In this work, we study both the ground-state and finite-temperature phase diagram of the long-range ferromagnetic quantum Ising model in one spatial dimension, by means of numerically exact, large-scale Path Integral Monte Carlo simulations. We perform our calculations for two representative values of $\alpha$: namely, we choose $\alpha = 0.05$ and $\alpha = 1.50$, within the extremely long-range region $\alpha < 1$ and intermediate region $1 < \alpha < 2$, respectively. We employ a wide variety of well-known finite-size scaling techniques to determine the position (i.e., the critical points) and critical exponents of both the ground-state and finite-temperature paramagnetic-ferromagnetic transitions displayed by the model, obtaining the phase diagram displayed in Fig. 1.

We determine the critical points and critical exponents for the ground-state ferromagnetic-paramagnetic transition. Our results for critical point positions and correlation length critical exponents are in agreement with existing predictions in the literature where the latter are available (i.e., $\alpha = 1.50$), while we encounter relatively small ($\sim 7\%$) deviations with respect to our estimate for the magnetization critical exponent. We then obtain accurate results for the position of the critical points in the finite-temperature regime for several values of the interaction strength. Concomitantly, our estimated correlation length critical exponents at $\alpha = 1.50$ essentially confirm the theoretical prediction of no qualitative deviations from the classical universality class due to quantum fluctuations, while discrepancies (up to 10% in the strongly interacting region) appear in the susceptibility critical exponent.

The structure of the paper is the following. Sec. 2 introduces the Hamiltonian, the numerical technique employed for its study, and the finite-size scaling approaches we employed to analyze its critical behavior. Sec. 3 discusses our obtained results on the critical behavior of the model. Finally, in Sec. 4 we outline the conclusions of our work and offer an outlook for future direction of research.

## 2 Model and methods

### 2.1 Hamiltonian and known results

The model analyzed in this work is described by the Hamiltonian

$$H = -\frac{V}{K(L)} \sum_{i<j} \frac{S_i^z S_j^z}{r_{ij}^\alpha} - h \sum_i S_i^x, \tag{1}$$

where $V > 0$ is the interaction strength, $i, j$ run over the sites $1, \dots, L$ of a one-dimensional lattice with periodic boundary conditions, $r_{ij}$ is the distance between sites $i$ and $j$, $S_i^z$ ($S_i^x$) is the component along $z$ ($x$) of the spin-1/2 operator acting on site $i$, and $K(L) \equiv (L-1)^{-1} \sum_{i \neq j} r_{ij}^{-\alpha}$ is the Kać renormalization factor. The latter ensures the existence of a proper thermodynamic limit in the regime $\alpha \leq 1$, while for $\alpha > 1$ it amounts to a rescaling of the interaction strength, and does not change the universal features of the critical behavior of the model. We remark that the presence of this renormalization factor

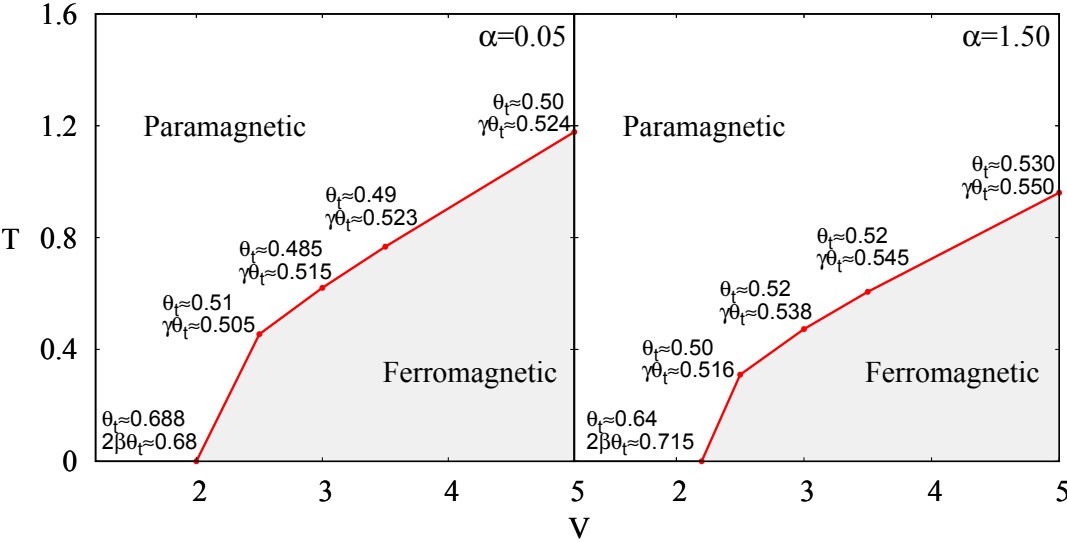

Figure 1: Calculated phase diagram of the long-range transverse-field Ising model in eq. (1), displaying the ground-state and finite-temperature phase boundary and critical exponents obtained using finite-size scaling techniques. Panels (a) and (b) correspond to $\alpha = 0.05$ and $\alpha = 1.50$, respectively. Here, $T$ is the system temperature in units of the Boltzmann constant, and $V$ is the interaction strength in units of the transverse field (see below). The displayed results for the effective thermal exponent and its product with the magnetization and susceptibility critical exponent are those obtained via data collapse (see below).

is directly related to how interactions with $\alpha < 3$ are engineered in trapped ions experiments. The latter exploit coupling between the ions and collective modes of the ion chain (phonons), mediated via a single laser shined over the full sample. Increasing the number of ions while keeping the lattice spacing constant naturally leads to a reduced coupling strength, that translates into the fact that the energy of the full system is still extensive - as reflected by Kać normalization. In the following, periodic boundary conditions are taken into account following the minimum-image convention, and $h = 1$ will be taken as unit of energy.

For very small interaction strength $V$, the ground state of the system in the thermodynamic limit is a paramagnet, characterized by a vanishing value of the magnetization along the $z$ direction $|m_z| \equiv L^{-1}|\sum_i S_i^z|$. On the contrary, for $V \gg 1$ the system is in a ferromagnetic phase, displaying a finite $|m_z|$. The existence of a finite-$V$ phase transition connecting these two states can be proven via analytical arguments (see, e.g., [17]); its UC depends strongly on the value of the decay parameter $\alpha$. Indeed, the $\alpha = 0$ case, also referred to as *Lipkin-Meshkov-Glick model* [37], can be described in an exact fashion at the mean-field level [38], and the paramagnetic-ferromagnetic transition has been proven to be of the mean-field type in the $1 < \alpha < 5/3$ region. In contrast, in the regime $\alpha \geq 3$, the critical point belongs to the short-range UC (i.e., the one of the ferromagnetic-paramagnetic transition in the nearest-neighbor limit $\alpha \to \infty$).

In the finite-temperature regime, generic scaling arguments [35] predict that the model should display the same critical behavior as its classical (i.e., $h = 0$) counterpart, due to the finiteness of the system size in the imaginary time dimension (see below). The critical behavior of the classical model has been studied via both analytical (see, e.g., [39]), RG (see, e.g., [40]) and numerical techniques (see, e.g., [41]) in the $\alpha > 1$ regime. Here, the

142  system displays a second-order ferromagnetic-paramagnetic thermal phase transition for
143  $1 < \alpha < 2$, with the region $1 < \alpha < 3/2$ belonging to the mean-field regime, while in the
144  point $\alpha = 2$ the model undergoes a finite-temperature transition of the BKT type, and
145  the short-range regime is reached (i.e., no finite-temperature transition takes place) for
146  $\alpha > 2$.

## 2.2 Numerical techniques and finite-size scaling

148  We perform our investigation of the Hamiltonian in eq. (1) via Path Integral Monte Carlo
149  (PIMC) [42], a numerically exact technique for the study of unfrustrated systems of bosons
150  and quantum spins. In this approach, one maps the features of a quantum model of
151  interest to those of an equivalent, higher-dimensional classical one, which is then studied
152  via Metropolis Monte Carlo simulations. The quantum-to-classical mapping described
153  above maps the partition function of the extended transverse-field Ising model in eq. (1)
154  into the one of an anisotropic extended Ising model on a rectangular lattice, via a procedure
155  known as *Suzuki-Trotter breakup*. Here, in addition to the original spatial dimension, one
156  also considers a discretized and periodic one, known as *imaginary time*, which extends
157  in the interval $[0, \beta]$, where $\beta = 1/T$ is the inverse system temperature in units of the
158  Boltzmann constant. The number of sites $M$ along this direction (also known as *slices*) is
159  a free parameter which affects the accuracy of the mapping: indeed, the latter is exact up
160  to $O(\beta/M)$ corrections, which vanish in the limit $M \to \infty$.

161  In the spatial direction, the extended Ising model resulting from the mapping displays
162  the same ferromagnetic long-range interactions present in the spin-spin term of the model
163  in eq. (1), while spin-spin couplings are nearest-neighbor in the imaginary time direc-
164  tion. Our PIMC algorithm combines conventional Wolff cluster updates [43] in imaginary
165  time with efficient long-range cluster updates [41] in the spatial direction. The choice of
166  these two state-of-the-art techniques allow to accurately analyze large system sizes (up
167  to $L = 8192$ sites) at low enough temperatures (down to $\beta = 1024$) to reach the ground
168  state regime. The Suzuki-Trotter corrections mentioned above are kept into account by
169  performing simulations with increasing number of slices (up to $M = 65536$), until a value
170  $M = M^*$ is found such that the corresponding values of the observables of interest were
171  determined to be identical, within statistical error, to those obtained for $M = 2M^*$. The
172  same protocol (with $\beta$ in the place of $M$) is adopted to ensure the $T \to 0$ limit is reached
173  in the investigation of the ground state regime.

174  The PIMC algorithm gives us direct access to observables commuting with the $S_i^z$
175  operators, including the integer powers of $|m_z|$. This allows us to compute quantities such
176  as the Binder cumulant

$$U = \frac{1}{2}\left[3 - \frac{\langle m_z^4 \rangle}{\langle m_z^2 \rangle^2}\right],\tag{2}$$

177  where $\langle \ldots \rangle$ stands for statistical averaging, which is expected to converge to 1 (0) in a
178  ferromagnetic (paramagnetic) phase [44]. We also compute the "classical" susceptibility

$$\chi = \beta L\left(\langle m_z^2 \rangle - \langle |m_z| \rangle^2\right),\tag{3}$$

179  which, in proximity of a finite-temperature critical point of a quantum model, approxi-
180  mates well the exact functional form of the magnetic susceptibility [36].

181  In order to extract reliable information on the critical behavior of the model in the
182  thermodynamic limit, we exploit the well known finite-size scaling (FSS) theory [44]. In
183  this framework, scaling relations of various quantities in terms of the correlation length
184  $\xi$, which diverges when approaching a critical point, are exploited to obtain finite-size
185  information by noting that in a finite system $\xi$ will saturate to a value $O(L)$, where $L$ is the

system size. Features such as the position of the critical point or the critical exponents, on which the original scaling relations depended, can then be directly extracted via numerical fits as a function of $L$. In the following section, when discussing the fitting procedures to obtain such quantities, we will offer detailed formulae regarding FSS predictions for observables such as $U$ and $\chi$.

# 3  Results

We investigate the critical properties of the model in eq. (1) in the ground-state and finite-temperature regime for $\alpha = 0.05$ and $\alpha = 1.50$.

## 3.1  Ground-state critical behavior

The first step in our analysis is the determination of the paramagnetic-ferromagnetic critical point $V_c$ in the ground-state regime, which we accomplish by fitting to our numerical data for the Binder cumulant $U$ its expected FSS behavior. The Binder cumulant curves $U(V)$ for system sizes $L$ and, e.g., $2L$ are expected to cross at size-dependent points $V = V_U(L)$, which will follow (to the leading order) the FSS scaling [23, 45]

$$V_U(L) = V_c \left( 1 + a L^{-\omega - \theta_t} \right), \tag{4}$$

where $V_c$ is the critical point, and the *effective thermal exponent* $\theta_t$ is linked to the correlation length critical exponent $\nu$.

In the ground-state regime $\nu^{-1} = \theta_t$ outside of the mean-field region; conversely, when the latter is entered, corrections to the leading scaling behavior can be taken into account [23] via the generalized expression $\nu^{-1} = (d_{\mathrm{uc}}(\sigma)/d)\,\theta_t$, where $d$ is the dimensionality and $d_{\mathrm{uc}}(\sigma) = 3\sigma/2$ is the upper critical dimension for the value of $\sigma$ of intererest.

Comparison of eq. (4) with the predicted leading-order FSS behavior for the *value* of the Binder cumulant at the $V_U(L)$s,

$$U(L, V_U(L)) = b + c L^{-\omega}, \tag{5}$$

allows us to obtain estimates for $V_c$ and $\theta_t$, by fitting our computed results for the crossing features [see Fig. 2(a)] with the functional forms above.

Fig. 2(b-c) display examples of the FSS fitting procedures mentioned above; the obtained values of the critical point and of the effective thermal exponent $\theta_t$ are listed in Table 1.

| $\alpha$ | $V_c$ (BC) | $V_c$ (DC) | $\theta_t$ (BC) | $\theta_t$ (DC) | $2\beta\theta_t$ (DC) |
|---|---|---|---|---|---|
| 0.05 | 1.9997(4) | 1.9999 | 0.50(7) | 0.688 | 0.68 |
| 1.50 | 2.1972(7) | 2.1981 | 0.39(6) | 0.64 | 0.715 |

Table 1:  Values of $V_c$, $\theta_t$, and $\beta_m$ (see text) associated to the ground state paramagnetic-ferromagnetic transition, computed via FSS analysis of the Binder cumulant crossings (BC) and via data collapse of the squared magnetization $m_z^2$ (DC).

In order to gain more insight into the ground-state critical behavior of the model, we perform a data collapse analysis by directly exploiting the FSS predictions for the behavior of the squared magnetization close to a critical point [23, 44],

$$m_z^2 \sim L^{-2\beta_m \theta_t} \cdot f \left[ L^{+\theta_t} \left( V_c - V \right) \right] \qquad V \gtrsim V_c, \tag{6}$$

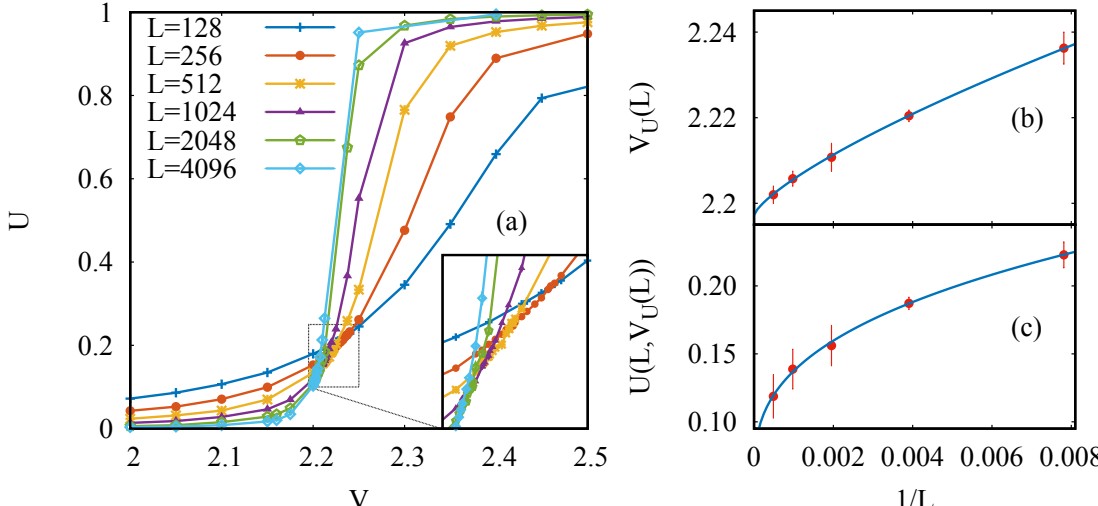

Figure 2: Binder cumulant scaling in the ground state regime (in all panels, $\alpha = 1.50$).
Panel (a): Binder cumulant curves as a function of $V$ for different system sizes. Solid
lines are a guide to the eye. Inset: magnification of the curve crossing region. Panel
(b): computed crossing positions $V_U(L)$ between the Binder cumulant curves at system
sizes $L$ and $2L$. The continuous line is a numerical fit to the expected FSS behavior in
eq. (4). Panel (c): computed values of the Binder cumulant at the crossing points $V_U(L)$
between system sizes $L$ and $2L$. The continuous line is a numerical fit to the predicted
FSS behavior in eq. (5).

216 where $\beta_m$ is the magnetization critical exponent, up to corrections of higher order in $1/L$.
217 This scaling law implies that the rescaled magnetization curves $y_L^m \equiv m_z^2(L)L^{+2\beta_m\theta_t}$ for
218 different system sizes should coincide if plotted as a function of $x_L^V \equiv (V_c - V)L^{\theta_t}$. We
219 perform a high-order polynomial fit of $y_L^m$ as a function of $x_L^V$ in a window around the
220 critical point $x_L^V = 0$ for a wide range of candidate values of $V_c$, $\theta_t$ and $\beta_m$, choosing
221 as our final estimates for these quantities the values which resulted in the fit with the
222 lowest chi-square value. While it is hard to assign a rigorous errorbar to the results of a
223 data collapse analysis, we estimate the order of magnitude of the error on our results by
224 performing the same fits in a considerably larger (i.e, containing of the order of double
225 the number of points) window around the critical point, and taking the difference between
226 the optimal values of $V_c$, $\theta_t$, and $\beta_m$ for the two windows as the order of their numerical
227 uncertainty.

228    Our collapsed data is displayed in Fig. 3(a-b); the obtained estimates for $V_c$, $\theta_t$ and
229 $\beta_m$ are listed in Table 1. We note that the data collapse behavior takes place over a fairly
230 wide range of values of the rescaled order parameter $x_L^V$, despite relatively narrow fitting
231 windows for the scaling behavior in eq. (6) (the intervals between dashed lines in Fig. 3).
232 This highlights the faithfulness of the data collapse scaling description of our numerical
233 data, which translates to highly reliable estimates of the critical properties of the system.

234    Examination of our results points out i) the remarkable agreement of the critical point
235 estimates obtained via the Binder cumulant FSS and the data collapse, and ii) conversely,
236 the incompatibility between the two estimates for the effective thermal exponent $\theta_t$. Due
237 to the arguments mentioned above, we believe the data collapse estimates for the critical
238 features to be more reliable in this regard.

239    For $\alpha = 1.50$, we find agreement for $\theta_t$ and deviations of the order of 7% for $2\beta\theta_t$ from
240 the independent SSE predictions in Ref. [23] which, in our notation, are $\theta_t \simeq 2\beta_m\theta_t \simeq$

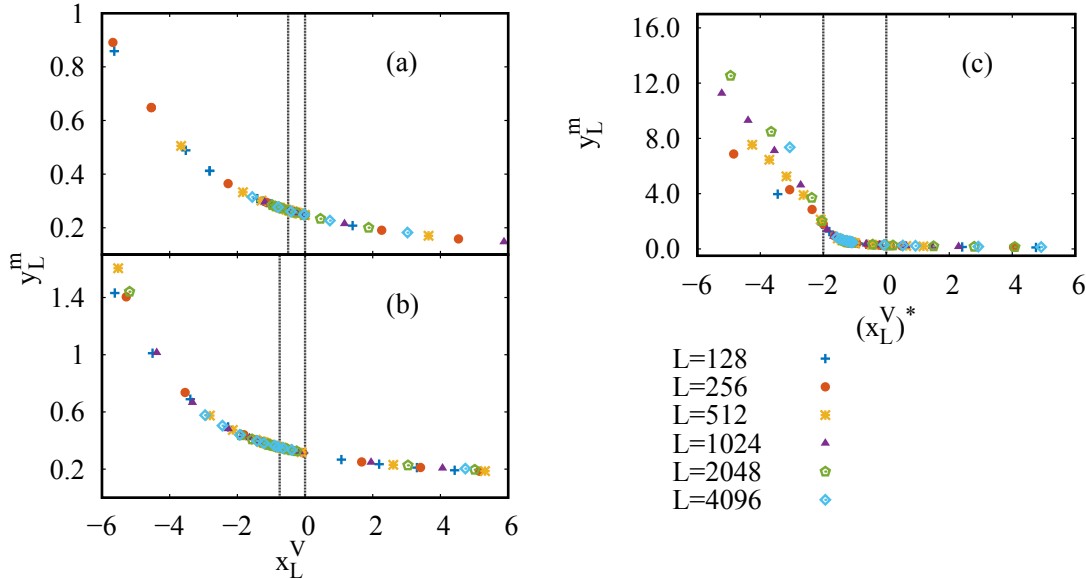

Figure 3: Panel (a): data collapse of the rescaled squared magnetization $y_L^m$ as a function of the rescaled interaction strength $x_L^V$ for $\alpha = 0.05$. Panel (b): same as panel (a) for $\alpha = 1.50$. Panel (c): same as panel (b), where the data collapse rescaling is performed on the Kać-factor-free rescaled interaction (see text). In all panels, the black dashed lines enclose the interval of the independent variable within which the data collapse scaling fit has been performed.

0.667. We also find good agreement with the estimate $V_c \simeq 0.42$ (in our notation) given in [23] for the position of the ground-state critical point, by performing a data collapse where the rescaled interaction $x_L^V$ is replaced by $\left(x_L^V\right)^* \equiv L^{+\theta_t}\left(V_c - V/K(L)\right)$ (the rescaling is required since the Kać correction factor is not employed in [23]). The resulting data collapse [see Fig. 3(c)] yields optimal values $\theta_t \simeq 0.64$, $2\beta\theta_t \simeq 0.76$, and $V_c \simeq 0.42$. For $\alpha = 0.05$, our estimates for $\theta_t$ and $2\beta\theta_t$ are compatible (up to deviations of the order of 3% in $\theta_t$) with the ones corresponding to the $\alpha = 0$ mean-field critical behavior, i.e., $\theta_t = 2\beta_m\theta_t = 2/3$ [38].

## 3.2 Finite-temperature critical behavior

Once the boundary of the ground-state ferromagnetic phase is determined, we investigate whether or not ferromagnetic order survives for $T > 0$, and more in general the details of the critical behavior of the model in this regime. To this end, we perform finite-temperature calculations for fixed values of $V$ belonging to the ferromagnetic phase in the ground state regime. We apply the FSS framework to quantities such as the Binder cumulant and the susceptibility, computed as a function of $T$, to estimate features of the temperature-driven critical behavior.

Indeed, our results for the Binder cumulant as a function of $\beta$ at fixed $V$ and different system sizes immediately confirm the presence of a finite-temperature phase transition, as pointed out by the appearance of the crossing behavior discussed above [see Fig. 4(a)] at size-dependent points $\beta_U(L, V)$. We determine the $V$-dependent critical temperatures $\beta_c(V)$ and the associated $\theta_t(V)$ via fitting of the FSS relations in eqs. (4)-(5) to our computed crossing features, with the thermal critical points $\beta_c$ and $\beta$ taking the role of $V_c$ and $V$, respectively. If the hypothesis of essentially classical critical behavior for the finite-temperature quantum model holds (as we argue below) one may link [46] $\theta_t$ to

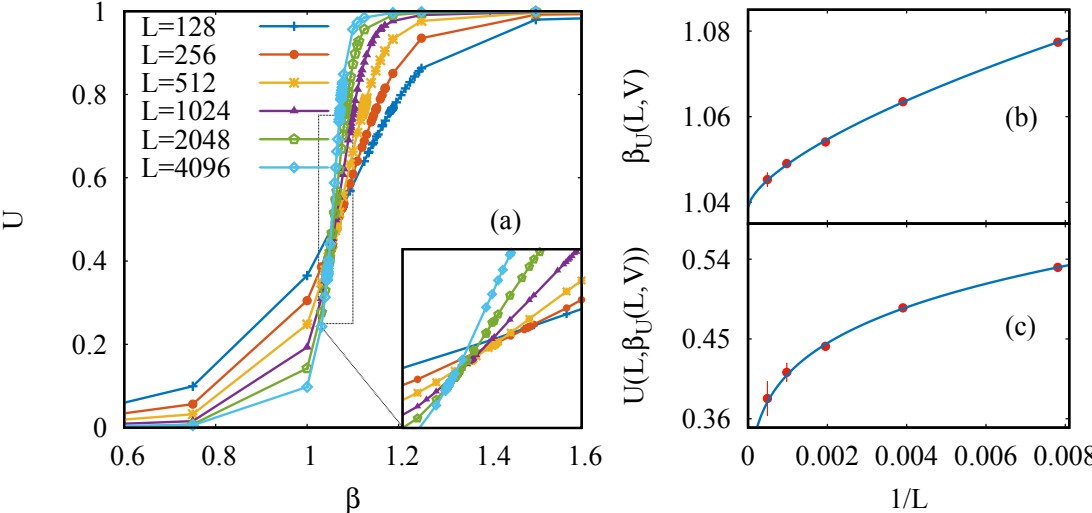

Figure 4: Binder cumulant scaling in the finite-temperature regime (in all panels, $\alpha = 1.50$ and $V = 5.0$). Panel (a): Binder cumulant curves as a function of $\beta$ for different system sizes. Solid lines are a guide to the eye. Inset: magnification of the curve crossing region. Panel (b): computed crossing positions $\beta_U(L, V)$ between the Binder cumulant curves at system sizes $L$ and $2L$. The continuous line is a numerical fit to the expected FSS behavior in eq. (4). Panel (c): computed values of the Binder cumulant at the crossing points $\beta_U(L, V)$ between system sizes $L$ and $2L$. The continuous line is a numerical fit to the predicted FSS behavior in eq. (5).

the correlation length critical exponent $\nu$ via the relation $\nu^{-1} = \left(d_{\mathrm{uc}}^{\mathrm{class}}(\sigma)/d\right)\theta_t$, where $d_{\mathrm{uc}}^{\mathrm{class}}(\sigma) = 2\sigma$ is the classical upper critical dimension.

Examples of this analysis are displayed in Fig. 4(b-c): the obtained critical parameters are listed in Table 2. We remark here that our application of this approach encountered in some cases strong difficulties due to significant finite-size effects in proximity of the $\beta_c(V, L)$. In particular, the relatively large numerical uncertainties on the values of the Binder cumulant in this region led to the necessity to perform conservative estimates of the finite-size crossing points. In turn, this prevented us in some cases from obtaining meaningful (i.e., with small enough errorbars) estimates for $\theta_t$.

In order to obtain an independent estimation of our quantities of interest, we investigate the finite-temperature behavior of the magnetic susceptibility for the same values of $V$ selected in our Binder cumulant analysis. At finite system size and fixed interaction strength, $\chi$ is expected to display peaks at size-dependent temperatures $\beta_\chi(L, V)$; the FSS framework predicts for the latter [23, 44] the leading scaling behavior

$$\beta_\chi(L, V) = \beta_c + fL^{-\theta_t} \tag{7}$$

as a function of the system size.

Our numerical data confirm the expected behavior of $\chi$ [see Fig. 5(a)]. Fitting the FSS functional form in eq. (7) to the computed peak positions [see Fig. 5(b) for an example] allows us to directly estimate the critical temperatures and effective thermal exponents as a function of the interaction strength (see Table 2 for a list of results).

While also requiring conservative estimates (and therefore large errorbars) for the peak positions, due to strong finite-size effects, we found the susceptibility-based approach to be much less sensitive to this issue than the Binder cumulant FSS discussed above. In particular, we encountered problematic results only for $V = 2.5$, for both values of $\alpha$

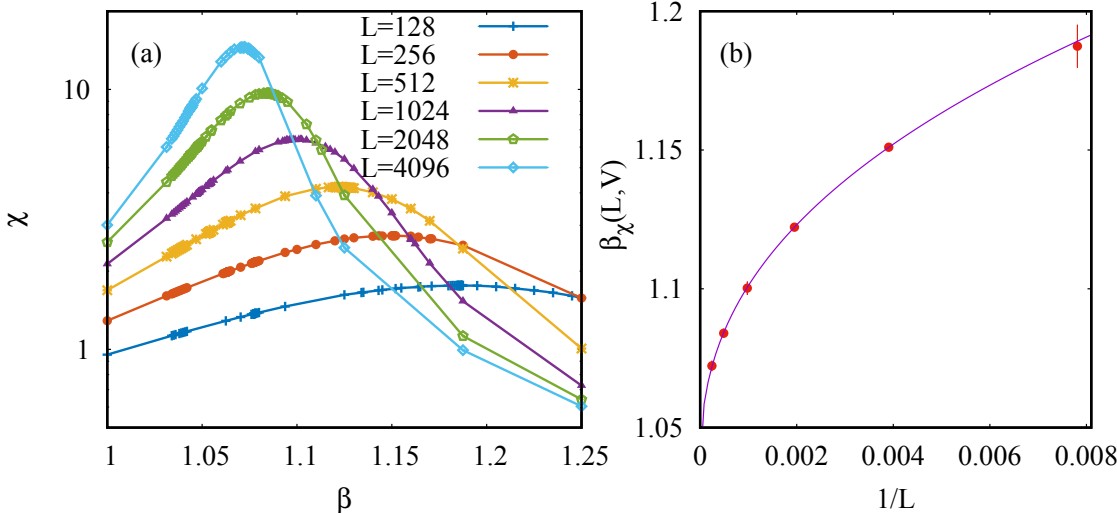

Figure 5: FSS analysis of the magnetic susceptibility in the finite-temperature regime (in all panels, $\alpha = 1.50$ and $V = 5.0$). Panel (a): susceptibility curves as a function of $\beta$ for different system sizes. Solid lines are a guide to the eye. Panel (b): finite-size peak positions $\beta_\chi(L)$. The continuous line is a numerical fit to the expected FSS behavior in eq. (7).

considered in this work, where our estimates were strongly dependent on the set of system sizes considered in the fitting procedure (the reported results correspond to the fits with all sizes considered).

We finally analyze the critical properties of the model by performing a data collapse analysis for the behavior of the magnetic susceptibility close to the finite-temperature critical points [23, 41, 44],

$$\chi \sim L^{+\gamma\theta_t} \cdot f\left[L^{+\theta_t}\left(\beta_c - \beta\right)\right] \qquad \beta \sim \beta_c, \tag{8}$$

where $\gamma$ is the susceptibility critical exponent, up to corrections of higher order in $1/L$. The analysis follows the same protocol outlined in our discussion of the ground-state regime, with the rescaled dependent and independent variables here being $y_L^\chi \equiv \chi(L)L^{-\gamma\theta_t}$ and $x_L^\beta \equiv (\beta_c - \beta)L^{\theta_t}$, respectively.

Fig. 6 displays our collapsed data for all the values of $\alpha$ and $V$ investigated in this work; the corresponding optimal (in the sense discussed above) results for $\beta_c$, $\theta_t$ and $\gamma$ are displayed in Table 2. As in the ground-state regime, we observe that the parameter range in which the data collapse scaling *ansatz* is respected noticeably exceeds our fitting window (and vastly so, in most cases), highlighting the accuracy of this approach in describing the critical behavior of the model. Furthermore, this protocol does not require the estimation of size-dependent features, sush as the curve crossings for the Binder cumulant, or the peak position for the susceptibility, allowing us to obtain much more reliable and systematics-free results. We also note that high degree of accuracy with which the scaling law in eq. (7) can be applied to describe the behavior of the "classical" susceptibility in eq. 3 is a strong indication of the goodness of the latter as an approximation for the complete functional form of the magnetic susceptibility.

A direct analysis of the results for the critical exponents listed in Table 2 shows that our estimates obtained via FSS of the Binder cumulant crossings, where meaningful in the sense discussed above, are consistent within errorbar with the ones obtained via suscepti- bility data collapse. Concomitantly, in some points we observe differences (which remain

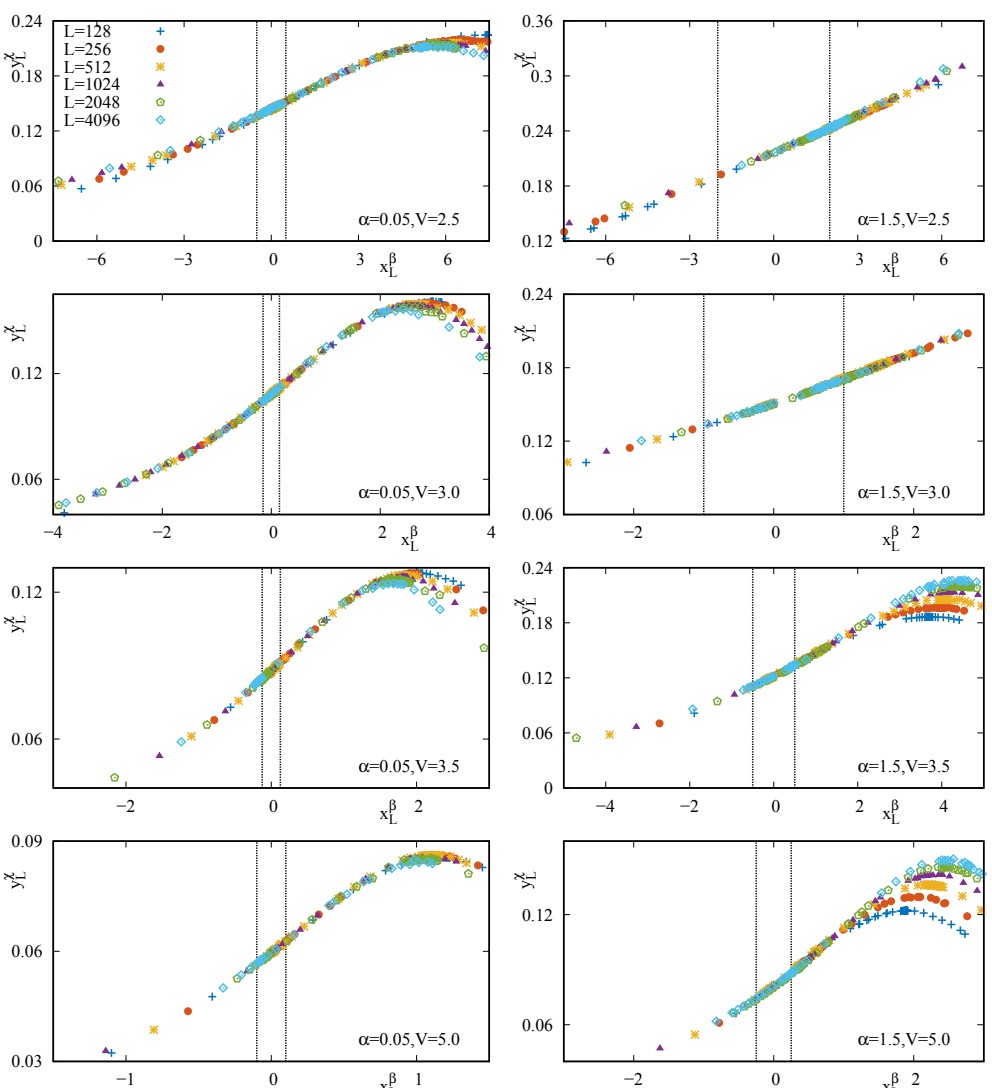

Figure 6: Data collapse of the rescaled magnetic susceptibility $y_L^\chi$ as a function of the rescaled order parameter $x_L^\beta$ for the values of $\alpha$ and $V$ studied in this work. The black dashed lines enclose the interval of $x_L^\beta$ within which the data collapse scaling fit has been performed.

consistently small, except for the point $\alpha = 1.50, V = 5.00$) between the latter and the results of the susceptibility peak position FSS for the values of $V$ in which the latter have converged with respect to the system sizes employed in the fitting procedure. In the points where this did not happen, the $\theta_t$ result from the susceptibility peak position fit decreased, shifting towards the data-collapse results, when smaller sizes were discarded.

According to the arguments mentioned in Sec. 2, the universality class of the $T > 0$ ferromagnetic-paramagnetic transition should be the same of the corresponding transition in the classical counterpart of model eq. (1). For $\alpha = 1.50$, the classical Hamiltonian is in the mean-field regime, and RG predictions, confirmed by classical Monte Carlo calculations [41], yield the estimates $\theta_t = \gamma\theta_t = 1/2$. Direct comparison with our most representative and reliable results in Table 2 (i.e., the one obtained via data collapse of the magnetic susceptibility) shows that our estimates for $\theta_t$ are in essential agreement with the classical prediction (with deviations outside of the estimated order of magnitude of the

| | | $\beta_c$ | | | $\theta_t$ | | | $\gamma\theta_t$ |
|---|---|---|---|---|---|---|---|---|
| | $V$ | $U$ | $\chi$ | $\chi_{dc}$ | $U$ | $\chi$ | $\chi_{dc}$ | $\chi_{dc}$ |
| $\alpha = 0.05$ | $V = 2.5$ | 2.2007(4) | 2.23(1) | 2.20 | / | 0.72(4)* | 0.51 | 0.505 |
| | $V = 3.0$ | 1.6120(7) | 1.61(1) | 1.612 | / | 0.54(3) | 0.485 | 0.515 |
| | $V = 3.5$ | 1.299(1) | 1.303(3) | 1.303 | / | 0.54(2) | 0.49 | 0.523 |
| | $V = 5.0$ | 0.8474(2)* | 0.844(2) | 0.8491 | 0.5(1) | 0.47(2) | 0.50 | 0.524 |
| $\alpha = 1.50$ | $V = 2.5$ | 3.21(1) | 3.351(9) | 3.229 | 0.49(7) | 0.75(1)* | 0.50 | 0.516 |
| | $V = 3.0$ | 2.109(1)* | 2.12(1) | 2.115 | 0.50(2) | 0.48(3) | 0.52 | 0.538 |
| | $V = 3.5$ | 1.647(6) | 1.646(5) | 1.650 | 0.5(2) | 0.46(2) | 0.52 | 0.545 |
| | $V = 5.0$ | 1.039(1) | 1.035(1) | 1.041 | 0.44(7) | 0.41(1) | 0.530 | 0.550 |

Table 2: Summary of the computed estimates for $\beta_c$, $\theta_t$, and $\gamma\theta_t$ (see text) for the finite-temperature transitions at our investigated values of $\alpha$ and $V$. Our results are categorized according to the methodology employed to derive them: namely, FSS of the Binder cumulant crossings ($U$), FSS of the magnetic susceptibility peak position ($\chi$), and data collapse of the susceptibility ($\chi_{dc}$). Estimates marked with an asterisk ($*$) did not converge with respect to the choice of minimum size to be included in the fitting procedure.

error only appearing for $V = 5.0$). Compatibility between our estimate and the theoretical predictions, even for $V = 5.0$, is confirmed by the results obtained via FSS of the Binder cumulant, while the susceptibility FSS estimates, where converged, show appreciable deviations only for $V = 5.0$. Conversely, our estimates for $\gamma\theta_t$ show relatively consistent deviations (up to the order of 10%), which increase with the interaction strength.

These differences with the predicted results may be in principle due to several causes, including i) the "classical" approximation employed for the study of the susceptibility in our analysis, or ii) genuine quantum effects which introduce deviations with respect to the predicted classical behavior. However, we find it unlikely that either (i) and/or (ii) may be the dominant physical mechanism underlying the observed deviations, since both effects are essentially quantum in nature, and are expected to become weaker for larger values of $V$, where in contrast our results are more at odds with the classically predicted values. Indeed, for higher interaction strengths quantum effects are expected to weaken, due to both the larger value of $V$ (in comparison to the transverse field $h$) and the higher temperature at which the critical region is located. This consideration leads us to the conclusion that despite these deviations (which may be caused by finite-size effects, or by higher-order corrections) the critical behavior of the model in this regime follows the classical UC.

As in the ground-state case, we find essential compatibility with the (classical) mean-field exponents at $\alpha = 0$; in particular, we match the predicted values [38] $\theta_t = \gamma\theta_t = 1/2$ up to relatively small deviations (of up to 2.5%) for the latter quantity, which also become larger in the strongly interacting regime, and are therefore likely not due to genuine quantum effects as argued above.

## 4 Conclusions and outlook

We study the ground-state and finite-temperature phase diagram and critical behavior of the long-range quantum Ising model in one spatial dimension, for values of the interaction exponent parameter of direct interest for current experiments in trapped ion setups. We perform numerically exact, large-scale PIMC simulations within both the extremely long-range region and intermediate long-range regime, respectively, employing a wide variety of

finite-size scaling techniques to determine the location (i.e., the critical points) and critical exponents of both the ground-state and finite-temperature phase transitions displayed by the model.

We determine transition points and critical exponents for the ground-state ferromagnetic-paramagnetic transition. We find essential agreement with existing predictions for these quantities, where available (up to small deviations for the value of the magnetization critical exponent), and compatibility of our extremely-long-range results with the fully-connected universal properties. We then accurately estimate the position of the critical points in the finite-temperature regime for several values of the interaction strength. Here, our estimated critical exponents in the intermediate-long-range region essentially confirm the theoretical prediction of classical universality. In particular, in the intermediate long-range regime our estimated correlation length critical exponent is fully consistent with the classical predictions, while the susceptibility one displays deviations at most up to the order of 10%. Similarly, in the extremely long-range region we find compatibility with the (classical) mean-field universality class up to deviations of the order of 2.5% in the value of the correlation length critical exponent.

Beyond exploring the equilibrium phase diagram and the nature of critical points, our work is also directly relevant for another open question appearing in the context of quantum Hamiltonians with long-ranged interactions. This concerns quantum thermalization and equilibration during coherent quantum dynamics without coupling to an environment, which appears all but settled. In the infinitely-connected limit of $\alpha = 0$ it is already well known that thermalization does not occur [47]. Furthermore, numerical works close to this infinitely-connected limit have already observed indications that thermalization could be prevented at least on the achievable time scales [48]. In order to settle this fundamental question, the understanding of the thermal equilibrium phases and properties, to which this work contributes, represents a first key step. While thermalization corresponds to ensemble equivalence of the thermal ensemble with the diagonal ensemble, capturing the long-time steady states during dynamics [49], it is also not known to which extent such long-range models exhibit ensemble equivalence on a general level. This concerns for instance the equivalence of the thermal and microcanonical ensemble, which is of central importance from the statistical physics point of view.

# Acknowledgements

We gratefully acknowledge discussions with K. Schmidt, A. Trombettoni, S. Ruffo, and A. Silva.

**Funding information** The work of AA, MD and EGL is partly supported by the ERC under grant number 758329 (AGEnTh), by the MIUR Programme FARE (MEPH), and has received funding from the European Union's Horizon 2020 research and innovation programme under grant agreement No 817482. This work has been carried out within the activities of TQT. This project has received funding from the European Research Council (ERC) under the European Unions Horizon 2020 research and innovation programme (grant agreement No. 853443), and MH further acknowledges support by the Deutsche Forschungsgemeinschaft via the Gottfried Wilhelm Leibniz Prize program.

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
