# Peer review of "Finite-temperature critical behavior of long-range quantum Ising models"

_SciPost Physics_

## Round 1 · Referee Report · Anonymous (Referee 1) · 2021-6-27

Strengths

  1. Deals with a very interesting model.

  2. Extensive numerical studies.

  3. Results are transparently clarified.

  4. Connection to thermalisation may lead to interesting future studies.

Weaknesses

Nothing worth mentioning. See the report below.

Report

The paper deals with a many interesting model namely the long-range ferromagnetic Ising model with power-law interactions decaying as $1/r^{\alpha}$. Both the ground state and thermal critical properties are studied using path integral Monte Carlo simulations. This is indeed a very interesting and long-standing problem and the authors comment on the connection with thermalisation. The authors have chosen two specific values of the range-parameter $\alpha$, one represents very long-ranged interactions while for the other vale the interaction lies within the intermediate range. The authors study finite size scaling schemes of the quantities like Binder cumulant and also verify the classical quantum correspondence in the context of critical exponents. Question is that whether the ground state ferromagnetic order survive at finite temperatures, if so associated finite temperatures phase transition and the critical exponents.

In short, this is a solid and thorough numerical study on a very important model which critically explores the ground state and finite temperature phase transitions. Especially, the studies of the region $\alpha <1$, are rare and quite intriguing. Further the discrepancy with expected critical exponents are explicitly clarified.

I therefore recommend the paper in the present form. I however have a few minor comments:

  1. In connection to the review of long-range interacting systems, the authors should cite "Journal of Physics A: Mathematical and Theoretical 53 (1), 013001"

  2. Authors should discuss one dimensional power-law interacting Kitaev chain Ref. [25], which is exactly solved, to a greater extent.

  3. The work focusses on Ising situation, authors may like to add a brief comment on the corresponding $n$-vector model?

With these minor changes, I recommend that the paper should be accepted for publication.

Requested changes

As suggested in the report above

  • validity: high
  • significance: high
  • originality: high
  • clarity: top
  • formatting: excellent
  • grammar: excellent

Author:  Eduardo Gonzalez Lazo  on 2021-08-11  [id 1656]

(in reply to Report 1 on 2021-06-27)

**Response:** We thank the Referee for the positive appraisal, and appreciate her/his recommendation for acceptance of our paper in SciPost. Below, we address the questions and comments raised in her/his report.

**Referee:**

>In connection to the review of long-range interacting systems, the authors should cite "Journal of Physics A: Mathematical and Theoretical 53 (1), 013001"

**Response:** We have added a citation to the mentioned review article at line 55.

**Referee:**

>Authors should discuss one dimensional power-law interacting Kitaev chain Ref. [25], which is exactly solved, to a greater extent.

**Response:** There is quite a sharp difference between the case we
investigate, and the long-range Kitaev model. In the new version of the
manuscript, we have (*i*) added a comment emphasizing differences between the two
cases, and (*ii*) pointed out a new element of the outlook – that is, that the
long-range Kitaev could represent another interesting playground to understand
symmetry-breaking transitions at finite temperature in the presence of
long-range couplings.

**Referee:**

>The work focusses on Ising situation, authors may like to add a brief comment on the corresponding n-vector model?

**Response: ** It is not immediately clear to us if our results could be informative about the transition properties of O(n) models. While some aspects (e.g., the nature of the transition line at T>0 for small alpha) could be similar, this has – to the best of our knowledge – not yet been investigated in detail. In view of this, we prefer to avoid possibly speculative comments (even if, of course, this is an interesting future research topic on its own).

---

## Round 1 · Referee Report · Anonymous (Referee 2) · 2021-7-1

Strengths

1) Timely topic 2) Strong numerical tool 3) Numerically challenging problem

Weaknesses

No major weak point.

Report

The article entiteled "Finite-temperature critical behavior of long-range quantum Ising models" by Gonzalez Lazo and collaborators investigates the paradigmatic quantum spin model with long-range interactions, the transverse-field Ising chain with algebraically decaying long-range Ising interactions, using quantum Monte Carlo simulations. The authors consider unfrustrated ferromagnetic Ising interactions so that the microscopic model is unfrustrated and several results concerning the critical properties are known from renormalization group calculations and other numerical calculcations, but mostly at zero temperature. The main focus of this paper are the finite-temperature critical properties of the model which are less studied. In my opinion the article is well written and the topic is interesting. Globally, I therefore recommend publication in SciPost. I nevertheless have some points, which the authors should address to further improve their manuscript.

Requested changes

1) Lines 92/359: The word "ferromagnetic-" extends the widths of the column. 2) Figure 1: I find the presentation of the all the values for theta and gamma*theta is not so nice yet. Maybe some more spaces would help. 3) Line 134: I would suggest to replace "mean-field type" by "Gaussian universality class" 4) Line 156: Why do the authors have used a discretized imaginary time and not an algorithm with continuous imaginary time? Is there a physical argument? 5) Lines 222/284: "errorbar" -> "error bar" 6) Line 368: "one displays" -> "displays"

  • validity: high
  • significance: high
  • originality: good
  • clarity: good
  • formatting: excellent
  • grammar: excellent

Author:  Eduardo Gonzalez Lazo  on 2021-08-11  [id 1657]

(in reply to Report 2 on 2021-07-01)

Response: We thank the Referee for the knowledgeable summary of our work, and appreciate her/his recommendation for acceptance of our paper in SciPost. Below, we address the questions and comments raised in her/his report.

Referee:

Lines 92/359: The word "ferromagnetic-" extends the widths of the column.

Response: We thank the Referee for pointing out this typo, that we fixed by defining appropriate, and much shorter, acronyms for the names of the model phases.

Referee:

Figure 1: I find the presentation of the all the values for theta and gamma theta is not so nice yet. Maybe some more spaces would help.

Response: We thank the Referee for pointing out this issue, of which we took care by increasing the spacing between the numerical values of the exponents in Fig. 1.

Referee:

Line 134: I would suggest to replace "mean-field type" by "Gaussian universality class"

Response: We thank the referee for this suggested replacement, with which we agree, and which we performed in the resubmitted version.

Referee:

Line 156: Why do the authors have used a discretized imaginary time and not an algorithm with continuous imaginary time? Is there a physical argument?

Response: While it is definitely possible to write path integral Monte Carlo (MC) algorithms in continuous imaginary time for transverse-field-Ising-type Hamiltonians [see, e.g., Blote et al., Phys. Rev. E 66 066110 (2002)], these have been proposed, to our knowledge, only in the case of nearest-neighbor interactions (generalizable to the case of short-range potentials). Combining the continuous-time data structure with the optimized cluster updates required to efficiently study long-range potentials proved a difficult task, leading us to choose the less efficient discrete-time algorithm in return for a more effective treatment of the spatial part of our Hamiltonian.

Stochastic Series Expansion (SSE) techniques [see, e.g., Sandvik, Phys. Rev. E 68 056701 (2003)] provide a powerful alternative to path integral MC approaches, and are free from imaginary-time discretization errors. As long as the latter are kept under control, as in our study, the two methods are essentially equivalent; the higher convenience and efficiency of SSE algorithms, however, has led us to strongly consider their adoption in future studies of this kind of Hamiltonian.

Referee:

Lines 222/284: "errorbar" -> "error bar"

Response: We thank the referee for this suggested replacement, with which we agree, and which we performed in the resubmitted version.

Referee:

Line 368: "one displays" -> "displays"

Response: We believe that the replacement suggested here by the Referee would suggest the deviations from existing predictions mentioned later in the sentence are for the susceptibility, while they are indeed for the associated critical exponent.

As such, we replaced "... the susceptibility one (critical exponent) displays..." with "... the susceptibility exponent displays...", to clarify the subject of the sentence, in the spirit of the Referee's suggestion.

---

## Editorial Decision

resubmitted